# Measuring the counterion cloud of soft microgels using SANS with contrast variation

Boyang Zhou [1,5], Urs Gasser [1,5] & Alberto Fernandez-Nieves [2,3,4]

The behavior of microgels and other soft, compressible colloids depends on particle concentration in ways that are absent in their hard-particulate counterparts. For instance, poly-*N*-isopropylacrylamide (pNIPAM) microgels can spontaneously deswell and reduce suspension polydispersity when concentrated enough. Despite the pNIPAM network in these microgels is neutral, the key to understanding this distinct behavior relies on the existence of peripheric charged groups, responsible for providing colloidal stability when deswollen, and the associated counterion cloud. When in close proximity, clouds of different particles overlap, effectively freeing the associated counterions, which are then able to exert an osmotic pressure that can potentially cause the microgels to decrease their size. Up to now, however, no direct measurement of such an ionic cloud exists, perhaps even also for hard colloids, where it is referred to as an electric double layer. Here, we use small-angle neutron scattering with contrast variation with different ions to isolate the change in the form factor directly related to the counterion cloud, and obtain its radius and width. Our results highlight that the modeling of microgel suspensions must unavoidably and explicitly consider the presence of this cloud, which exists for nearly all microgels synthesized today.

Soft and deformable colloids exhibit a phase behavior that is more complex than that of hard and incompressible colloids. Furthermore, there is no generally accepted model for their interactions and overall behavior covering both dilute and concentrated conditions. Microgels, cross-linked polymer particles immersed in a solvent, are an important class of soft colloids that is of interest for applications and for fundamental studies aimed at, for example, understanding the role of colloidal softness. It is in fact this softness that renders microgels responsive to changes in variables like temperature[1,2], pH[3,4], and hydrostatic pressure[5–7]. Their shape, size, internal architecture, and, as a consequence, their particle–particle interactions, all depend on the conditions of the surroundings.

The importance of particle softness was highlighted by the spontaneous deswelling of large particles in bidisperse suspensions of poly-*N*-isopropylacrylamide (pNIPAM) microgels suspended in water.

With increasing concentration, the large and less numerous microgels were found to spontaneously deswell to about the size of the smaller and more numerous microgels[8]. This reduced the bidispersity of the suspension, a behavior not known for other colloids or particulate materials, allowing crystallization in the absence of point defects. For pNIPAM microgels, deswelling can start at concentrations below random close packing, i.e., without direct contact between particles[9–11]. We previously identified the fixed charges at the periphery of the particles and their associated counterions to be the key to understand this spontaneous deswelling[10,12,13]. Charged -O-SO₂-O⁻ groups originating from the initiator of the polymerization reaction remain at the ends of pNIPAM chains and locate at the microgel outskirts Supplementary Information; their counterions thus form a cloud surrounding the microgel. Most of these counterions are electrostatically bound to the particle, but there is always a number of them

[1]Laboratory for Neutron Scattering and Imaging, Paul Scherrer Institut, Forschungsstrasse 111, 5232 Villigen, Switzerland. [2]Department of Condensed Matter Physics, University of Barcelona, Carrer de Martí i Franqués 1, Barcelona 08028, Spain. [3]ICREA-Institucio Catalana de Recerca i Estudis Avançats, Barcelona 08028, Spain. [4]Institute for Complex Systems (UBICS), University of Barcelona, Barcelona 08028, Spain. [5]These authors contributed equally: Boyang Zhou and Urs Gasser. ✉e-mail: urs.gasser@psi.ch

that, for entropic reasons, overcome this electrostatic attraction and wander around contributing to the suspension osmotic pressure. Deswelling occurs when the distribution of counterions inside and outside the microgel causes an osmotic-pressure difference $\Delta\pi$ comparable to the microgel bulk modulus. This can happen when the counterion clouds percolate through the accessible volume, causing a steep increase in $\Delta\pi$, as counterions that were previously bound to a particle become effectively free. We have shown that this mechanism can explain both the observed spontaneous deswelling[10] and also the concentration-dependent freezing transition in polydisperse microgel suspensions with either a bimodal or a Gaussian size distribution[11].

The distribution of fixed charges and the associated counterion cloud in an otherwise totally charge-neutral pNIPAM network is, therefore, of utmost importance to model and understand the deswelling behavior and the interaction between pNIPAM microgels. As the amount of fixed charge is related to particle synthesis, one can envision synthesizing microgels with tailored interactions and thus phase behavior. Up to now, however, no direct measurement of this cloud exists. Since the counterion density in this cloud is expected to provide a small contribution to the microgel form factor, directly proving its existence is challenging. In this paper, we use small-angle neutron scattering (SANS) to reveal the signal due to the counterion cloud by taking the difference between two samples that contain the same microgels at eseentially the same number densities but in the presence of different counterions, sodium (Na$^+$) or ammonium (NH$_4^+$). Suspensions containing either ion can be obtained using dialysis Supplementary Information. The contrast difference for neutron radiation between these two ions is the key that allows obtaining the signal due to the counterion cloud. Note that this cloud exists as an electric double layer in charged hard colloids. However, in this case, the particle rigidity prevents the suspension polydispersity from changing. For microgels, this change can happen due to their softness, highlighting the role that these ions can have in the suspension behavior. To our knowledge, however, no direct measurement of the counterion cloud exists for soft colloids. Our work with SANS thus opens the door for exploring the details of ion clouds in any charged colloidal species.

## Results

We study three temperature-sensitive pNIPAM microgels with a lower critical solution temperature (LCST) $T_c \approx 32\,°C$ in water. Their swollen

radii determined by SANS are $(85 \pm 1.1)$ nm (s1), $(124.3 \pm 1.2)$ nm (s2), and $(135.6 \pm 2.6)$ nm (s3). They all contain 2.0 mol% of crosslinker $N,N'$-methylene-bis-acrylamide (BIS) and are synthesized with ammonium persulfate (APS) as initiator. Besides the temperature, the suspension behavior is controlled by the particle volume fraction $\phi$. Since microgels can deswell, deform, and interpenetrate, $\phi$ is hard to know at high concentrations. Therefore, we use a generalized volume fraction $\zeta$, defined in terms of the dilute microgel size Supplementary Information. Consequently for dilute suspensions $\phi = \zeta$, while at higher concentrations, $\zeta > \phi$, reflecting deswelling, shape change, or interpenetration. To learn about the counterion cloud of the pNIPAM microgels, we prepare pairs of samples containing the same microgels at essentially the same $\zeta$, but with Na$^+$ or NH$_4^+$ as counterions. We achieve this by dialyzing against a NaCl or NH$_4$Cl solution at a high concentration, $(167 \pm 5)$ mM in our case; this guarantees the positive ion is the wanted one. The process, however, results in excess salt, which we remove by dialyzing again but against pure H$_2$O Supplementary Information. The suspensions are then freeze-dried and resuspended in D$_2$O for SANS measurements. We prepare pairs of samples at $0.1 \lesssim \zeta \lesssim 0.2$ to investigate microgels in relatively dilute states and at $0.48 \lesssim \zeta \lesssim 0.55$, where the percolation of counterion clouds is expected[10,12].

SANS measurements taken in the swollen $[T = (20\pm0.5)\,°C]$ and deswollen $[(45 \pm 0.5)\,°C]$ states for a representative sample pair are shown in Fig. 1. Recall that the scattering is dominated by the particles and that the counterions only provide a second-order contribution to the scattered intensity $I(q)$. Similarly, any small changes in the microgel structure, as induced by the ions, would also result in second-order effects. However, studies of the Hofmeister series have shown that the swelling behavior of pNIPAM is only affected and becomes dependent on the ion type at salt concentrations $\gtrsim 300$ mM. In the suspensions studied here, the ion concentration is at least ten times lower. Therefore, the pNIPAM structure is not expected to change with the exchange of counterions[14,15]. We then analyze our SANS measurements considering that the ions do not significantly change the scattering of the microgel particles and that they only weakly contribute to the form factor. Our data analysis is based on a model that either directly includes the signal due to the ion cloud, or indirectly accounts for the effect of the ions through a change in the structure of the polymer network. In the following, we present our analysis considering that the signal directly results from the ion cloud, and later discuss the alternative possibility that the detected signal is caused by the ions but that it manifests as a small change in the form factor of the polymer network.

We use accepted models for pNIPAM microgels, the fuzzy-sphere form factor[16], and the Percus−Yevick structure factor, including polydispersity[17–19]. For each data pair, we extract the radius of the particle core $R_c$, the associated polydispersity $\sigma$, and the width of the fuzzy microgel corona $\sigma_s$; see Supplementary Table 2 Supplementary Information. To correct for small differences in particle number density $n_d$ between samples with Na$^+$ and NH$_4^+$ ions, which is essential to extract the weak scattering signal due to the counterion cloud, we consider that, except for a small signal due to the counterion cloud, $I(q)/n_d$ should not depend on the ion type. Thus, we introduce the corrected scattering signal as $I_{corr}^{Na^+}(q) = I^{Na^+}(q)\,n_d^{NH_4^+}/n_d^{Na^+}$. This corrected Na$^+$ curve is hard to distinguish from the uncorrected curves, confirming that $n_d$ is indeed very close in every sample pair; see Supplementary Fig. 3.

To explicitly consider the signal due to the counterion cloud, we divide the scattering amplitude of a microgel into the contributions due to the polymer network and the signal caused by the ion cloud: $F^X(q) = F_p(q) + F_{ic}^X(q)$, where the superscript $X$ refers to the ion type, either Na$^+$ or NH$_4^+$. Note that the term due to the polymer does not depend on the ion type, as argued in section 5 of the Supplementary Information, and that the scattered intensity $I^X(q) \propto |F^X(q)|^2$. We then take the difference of the scattered intensities in the presence of Na$^+$ and NH$_4^+$ ions, and obtain $\Delta I_{corr}(q) = I_{corr}^{Na^+}(q) - I^{NH_4^+}(q) \approx$

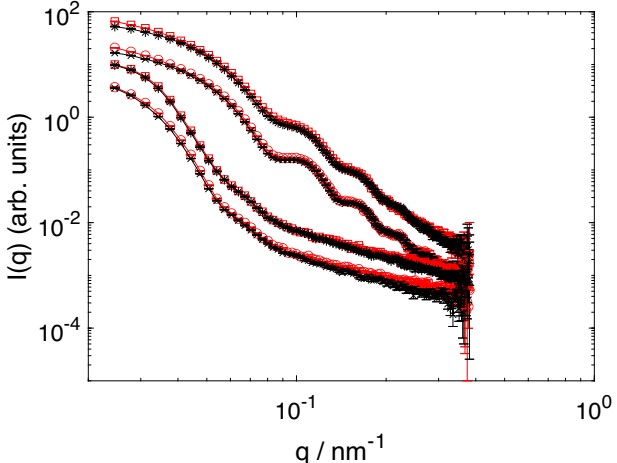

**Fig. 1 | SANS data of sample s3 with Na$^+$ (red circles and squares) and NH$_4^+$ counterions (black crosses and asterisks) taken at $(45 \pm 0.5)\,°C$ (two upper curves) and at $(20 \pm 0.5)\,°C$ (two lower curves).** For each counterion, two samples with concentrations $\zeta \approx 0.18$ (circles, crosses) and $\zeta \approx 0.54$ (squares, asterisks) were measured. The error bars represent the uncertainty due to the counting statistics of the SANS instrument.

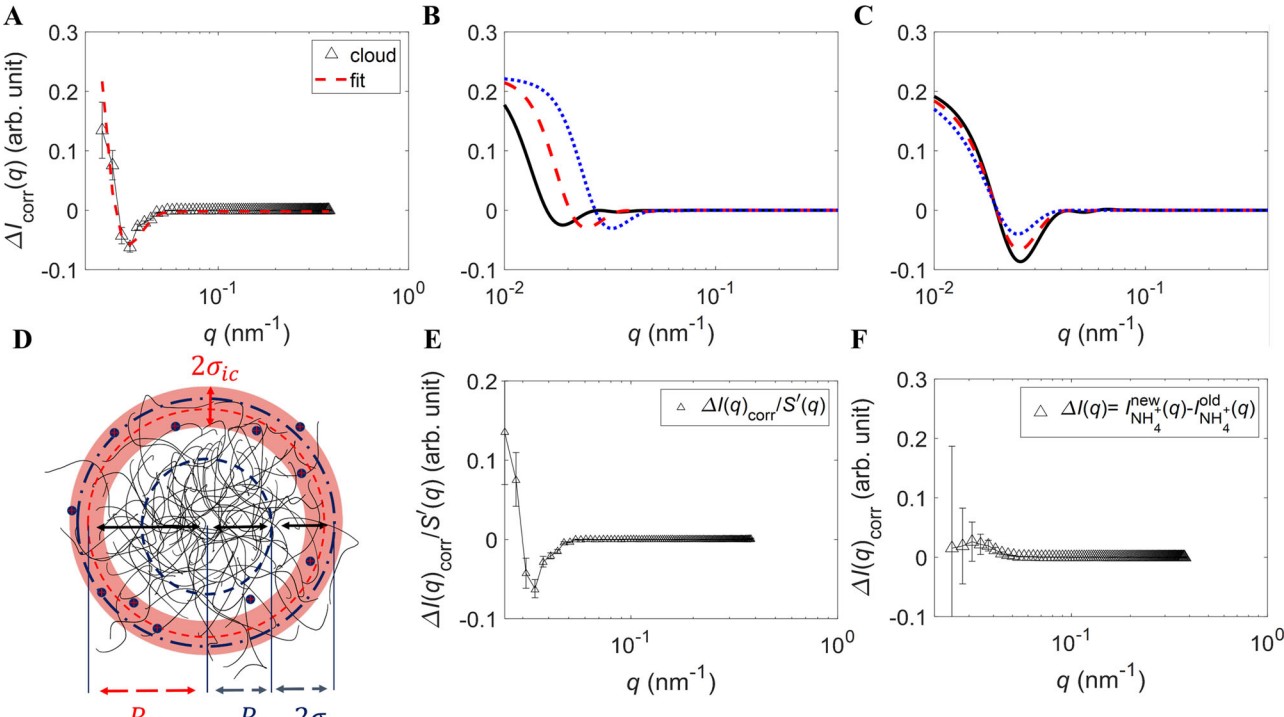

**Fig. 2 | SANS signal due to the counterion cloud. A** Subtraction $I^{NH_4^+}(q) - I_{corr}^{Na^+}(q)$ of sample s3 at $\zeta \approx 0.2$ and $T \approx 20\,°C$. The cloud fit is shown by the red, dashed curve. **B, C** Counterion cloud model including $S'(q)$, polydispersity, and instrument resolution for **B** $R_{ic} = R_{SANS} = 90$ nm (blue, dotted curve), 130 nm (red, dashed curve), 180 nm (black curve) with fixed parameters $\sigma_{ic} = 30$ nm, $\sigma_s = 15$ nm, $\phi = 0.2$, and $\sigma = 0.05$. **C** Counterion cloud model for $\sigma_{ic} = 50$ nm (blue, dotted curve), 30 nm (red, dashed curve), 10 nm (black curve) with fixed parameters $R_{ic} = R_{SANS} = 130$ nm,

$R_c = 100$ nm, $\sigma_s = 15$ nm, $\phi = 0.2$, and $\sigma = 0.05$. **D** Sketch of a pNIPAM microgel with $R_{SANS} = R_c + 2\sigma_s$ and the counterion cloud with width $2\sigma_{ic}$ and radius $R_{ic}$ located at the particle periphery. **E** Data shown in panel A divided by $S'(q)$. **F** Direct subtraction $I_{old}^{NH_4^+}(q) - I_{new}^{NH_4^+}(q)$ at $\zeta \approx 0.2$, $T \approx 20\,°C$, and with corrected number density. The error bars in panels **A**, **E**, and **F** represent the uncertainty due to the counting statistics of the SANS instrument.

$2F_p(q)\left[F_{ic}^{NH_4^+}(q) - F_{ic}^{Na^+}(q)\right]$, where we have neglected higher-order terms $\propto |F_{ic}^X(q)|^2$. We then consider the pre-factor due to the scattering contrast, $\Delta\rho^X$, and the volume of ions, $V^X$, and write $F_{ic}^X(q) = \Delta\rho^X V^X f_{ic}(q)$, where $f_{ic}(q)$ no longer depends on the ion type. This reveals that $\Delta I_{corr}(q)$ is directly proportional to the scattering amplitude of the cloud:

$$\Delta I_{corr}(q) \approx 2F_p(q)\left(\Delta\rho^{NH_4^+} V^{NH_4^+} - \Delta\rho^{Na^+} V^{Na^+}\right)f_{ic}(q) \quad (1)$$

We thus exploit the large contrast difference between the ions, $\Delta\rho^{Na^+} - \Delta\rho^{NH_4^+} \approx 9.8 \times 10^{-4}$ nm$^{-2}$, as well as the large scattering amplitude of the polymer network, to amplify the ion signal, which is shown in Fig. 2A.

We model the counterion cloud as a spherical surface that is smeared out due to the mobility of the ions and the arrangement of the fixed charges in the pNIPAM network. In reciprocal space we have

$$F_{ic}^X(q) = \Delta\rho^X V^X \frac{\sin(qR_{ic})}{qR_{ic}} \exp\left(-\frac{[q\sigma_{ic}]^2}{2}\right) \quad (2)$$

where $R_{ic}$ is the distance from the microgel center to the spherical counterion cloud, and $\sigma_{ic}$ quantifies the width of the cloud. Additionally, the model for the scattered intensity includes the structure factor of the suspension $S(q)$, the polydispersity, and the resolution of the SANS instrument Supplementary Information. This model reproduces the measured signal for all investigated samples, as shown by the dashed curve in Fig. 2A, and also in Supplementary Fig. 2 Supplementary Information, which is a fit to the data with $R_{ic}$ and $\sigma_{ic}$ as fitting parameters. Note that the detected cloud signal has the expected intensity. We quantitatively confirm this by taking the ratio of the measured intensities at $q = 0$, obtained from an extrapolation of

our fits, to obtain $\Delta I_{corr}(0)/I_{corr}(0) = 0.06 \pm 0.01$, which we compare with the corresponding ratio of the calculated prefactors, $s_{ic} = 2V_p\Delta\rho_p(\Delta\rho^{NH_4^+} V^{NH_4^+} - \Delta\rho^{Na^+} V^{Na^+})$, with $\Delta\rho_p = \rho_p - \rho_{D_2O}$ the contrast of the polymer network, and $s_p = (V_p\Delta\rho_p)^2$; we find $s_{ic}/s_p = 0.05 \pm 0.02$, where the error reflects the significant uncertainty in the number of counterions per microgel entering in $V^{NH_4^+}$ and $V^{Na^+}$ Supplementary Information. The agreement is very good, indicating that the observed signal is directly caused by the counterion cloud.

The fingerprint of the counterion cloud in $\Delta I_{corr}(q)$ is the presence of a negative minimum and an associated steep upturn at lower $q$. The $q$-position of the minimum is most sensitive to the cloud radius, and its depth is directly related to the width of the cloud, as illustrated in Fig. 2B, C.

The radii and widths of the whole microgel, obtained from the fits to the data shown in Fig. 1, as well as those for the counterion cloud, are shown in Figs. 3 and Supplementary Fig. 5 and are listed in Supplementary Table 2. For all investigated samples at $T \approx 20\,°C$, we find $R_{ic}$ to be smaller but close to $R_{SANS}$, indicating the counterion cloud is located inside the fuzzy shell of the microgel; see schematic in Fig. 2D. This confirms that the -O-SO$_2$-O$^-$ groups are indeed located in the particle periphery, as expected from the synthesis and in agreement with our model for particle deswelling at high concentrations[10,12]. In addition, the extent of the counterion cloud $\sigma_{ic}$ is on the order of the thickness of the fuzzy corona $2\sigma_s$. This is in good agreement with a prior estimate of the counterion cloud width, $(35 \pm 4)$ nm, obtained from the onset of microgel deswelling in bidisperse pNIPAM suspensions[11]. In that study, the freezing point, $\zeta_f$, was found to increase with the bidispersity of the suspension. As the bidispersity of the fully swollen microgels was too large for crystallization to occur, the formation of crystals relied on the deswelling of the largest and softest microgels and the corresponding

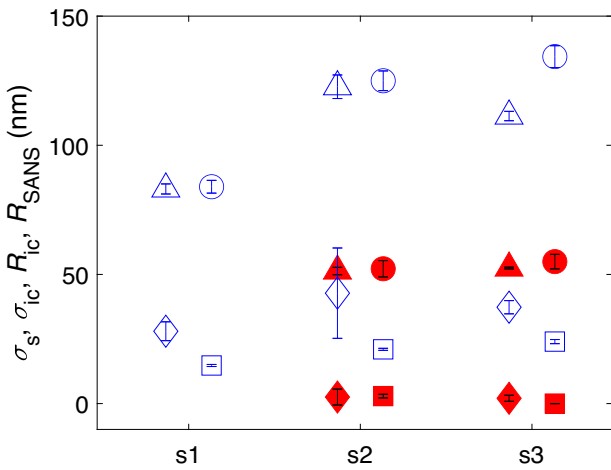

**Fig. 3 | Radii and shell widths of microgels and counterion clouds.** The radii $R_{SANS}$ (circles) and $R_{ic}$ (triangles) as well as the half widths of the fuzzy corona $\sigma_s$ (squares) and the counterion cloud $\sigma_{ic}$ (diamonds) for samples s1, s2, and s3 at $0.1 \lesssim \zeta \lesssim 0.2$ measured at $T \approx 20\,°C$ (blue open symbols) and $T \approx 45\,°C$ (red full symbols). All values shown are also listed in Supplementary Tables 2 and 3. The error bars result from the least-squares fit to the SANS data; see text for discussion.

reduction in bidispersity. Therefore, $\zeta_f$ could be used as a measure of the onset of deswelling due to the percolation of the counterion clouds and allowed for estimating its extent. Note $\sigma_{ic}$ is larger than the Debye screening length, which is $\kappa^{-1} \approx 14$ nm, due to the counterions in a microgel suspension at $\zeta = 0.5$. This suggests that, in the swollen state, the $-O\text{-}SO_2\text{-}O^-$ groups are distributed in the outskirts of the microgel's fuzzy corona and that this distribution increases the width of the counterion cloud.

While the microgels are fully swollen at $T = 20\,°C$ and $\zeta = 0.2$, their diameter is reduced by $\approx 4\%$ at $\zeta = 0.5$. If we consider an effective microgel including the counterion cloud, and calculate the volume fraction of these effective particles, we obtain $\zeta_{eff} = \zeta \left( \frac{R_{sw} + \sigma_{ic}}{R_{sw}} \right)^3 \approx 1.15$ for $\zeta = 0.5$, where $R_{sw}$ is the fully swollen microgel radius obtained with SANS. The suspension is, therefore, overpacked with these effective microgels and the counterion clouds must have percolated through the sample. The associated ions would then contribute to the suspension osmotic pressure, causing the observed deswelling of the particles. We note that the counterion-cloud model is expected to remain valid at $\zeta \approx 0.5$, since the increase in osmotic pressure is caused by a relatively small fraction of the ions in the counterion clouds. The associated deswelling of the microgels with increasing $\zeta$ then keeps the surface-to-surface distance between nearest neighbors, $d_{nn}$, larger than $\kappa^{-1}$; for $\zeta \approx 0.5$, we find $d_{nn} \approx 38$ nm, which is clearly larger than the Debye screening length $\kappa^{-1} \approx 14$ nm for sample s1 at $\zeta \approx 0.5$. Thus, most of the counterions remain in the spherical shell surrounding the microgels even at $\zeta \approx 0.5$.

We always find that the counterion cloud shrinks with the pNIPAM network. For measurements at $T \approx 45\,°C$, where the microgels are in the deswollen state, the core-shell structure disappears and the pNIPAM network essentially becomes a spherical particle with approximately constant polymer density. As illustrated by the filled symbols in Fig. 3, the counterion cloud is narrower and centered essentially at the particle surface, as expected if the fuzzy corona shrinks with the microgel, which should occur at temperatures above $T_c$. In this deswollen state, the same behavior is observed for both $\zeta \approx 0.2$ and 0.5, since, at this high $T$, the microgel bulk modulus is significantly higher than at 20 °C and $\zeta_{eff} < 1$ due to the reduced particle volume in the deswollen state; hence no counterion-induced deswelling is expected in this case. The width of the cloud decreases by a factor of 10 to 20 due to deswelling. Note that despite the error in $\Delta I_{corr}(q)$, associated with incoherent

scattering and instrument-specific backgrounds, the depth of the minimum is larger than the error for samples s1 and s3 at $T \approx 20\,°C$, see Supplementary Fig. 2 Supplementary Information, and that $\sigma_{ic}$ is observed to be strongly correlated to $2\sigma_s$.

To confirm our results, we perform additional tests. First, we confirm that the observed minimum is not caused by the structure factor of the suspension. We divide both $I^{NH_4^+}(q)$ and $I^{Na^+}(q)$ by their structure factors and determine $\Delta I_{corr}(q)$ after this operation. The minimum and the low-$q$ upturn in $\Delta I_{corr}(q)$ are still present, as shown in Fig. 2E. Furthermore, we obtain the same cloud parameters within the accuracy of the analysis. Second, we prepare a new sample with $NH_4^+$ counterions at $\zeta \approx 0.5$ and measure it at $T \approx 20\,°C$ to compare the result with the previously prepared $NH_4^+$ sample. Since the two samples both contain $NH_4^+$ counterions, data subtraction should not allow extracting the signal due to the counterion cloud, as there is no contrast difference between the ions. We follow the procedure described above to extract the counterion cloud and, after correcting for the possible slight difference in the particle number density, we find that the difference $\Delta I(q)$, shown in Fig. 2F, has neither a negative minimum nor an increased intensity upturn at low $q$ within the error of the measurements. No difference between the two samples is detected, corroborating the validity of our approach to extract the counterion cloud parameters. When we use the newly prepared sample with $NH_4^+$ counterions together with the sample containing $Na^+$ counterions, we can again detect the expected signal due to the counterion cloud and obtain the same cloud parameters as before within the accuracy of the analysis (see Supplementary Fig. 4) Supplementary Information.

We now address whether the fingerprint signal of the counterion cloud presented above could be reproduced without taking the counterions into account but by assuming instead a small structural difference in the pNIPAM network depending on the type of counterion. We had previously mentioned that this might perhaps be expected if the counterion concentration in the ion cloud was higher than what it actually is for our microgels. We nevertheless consider this possibility. If the signal shown in Fig. 2A is due to a change in the polymer network, the spherical shell model given by Equation (2) can still be applied for its analysis. The radii and widths listed above for the counterion cloud, however, now characterize the volume of the polymer network that is affected by exchanging the counterion. With this interpretation, the observed signal is still a consequence of the counterion cloud, but it is an indirect and not a direct measurement of the cloud. We now assume that the fuzzy-sphere model for the microgel form factor is accurate enough to model a change in the polymer network caused by the counterions and use it to try to explain the signal shown in Fig. 2A. As the change in the form factor happens close to the particle radius, the width of the fuzzy corona must be affected. We thus keep the core radius, $R_c$, fixed and consider the effects of small changes in $\sigma_s$. This approach can result in a negative minimum with the observed depth, but the width of the minimum is broader than in the measured signal. Hence the fuzzy-sphere model cannot capture the low-$q$ upturn that is otherwise naturally explained with the counterion-cloud model presented above, see Supplementary Fig. 6 Supplementary Information. A change in the polymer form factor must, therefore, be modeled with a contribution that goes beyond the fuzzy-sphere model and that is analogous to the cloud model given by Equation (2).

## Discussion
Our results emphasize the power of SANS with contrast variation to extract weak signals of interest. We are not aware of another method that has allowed extracting the spatial structure of an ion cloud with a density as low as that in the studied microgels. To put our analysis in perspective, we compare with previous cloud measurements for micellar systems[20]. In micelles, the charged head groups of the lipids or surfactants are arranged in a thin spherical shell that is not wider than 1 nm and contributes a

charge density of ~0.7$e$ nm$^{-3}$[20]. In contrast, the charged groups in a fully swollen pNIPAM microgel are distributed in the fuzzy periphery in a spherical shell with a thickness close to $\sigma_s$, which results in a charge density of ~0.015$e$ nm$^{-3}$, which is significantly lower than that for a micellar system. As the counterions arrange around the fixed charges on the particles, the counterions are spread out and have very low charge density. This low ion density explains the inability of anomalous small-angle x-ray scattering (ASAXS), a method that can extract the signal due to specific elements in the sample, to detect the counterion cloud. Other methods were also, in principle, applicable to our purposes. Ellipsometric light scattering (ELS), which can resolve structures close to the surface of colloidal particles[21], is an example. However, since this technique requires transparent samples, it could not be used with our concentrated, turbid suspensions. A second example is transmission-electron microscopy (TEM) with energy-dispersive x-ray spectroscopy (EDX), which allows for extracting the chemical composition of a sample. However, the ion density in our microgel suspension is too low for this technique to work. Additionally, the need to work with frozen samples makes it difficult to distinguish samples in swollen and deswollen states.

In the literature, the structure of counterion clouds has been reported for micellar systems by means of SANS[20,22,23], the combination of SANS and SAXS[24], and by means of ELS[21]. For the study of the coil-to-globule transition of DNA and sulfonated polystyrene, SANS was used to determine the ion distribution around the macromolecules[25]. The ion density in the counterion clouds in all these systems is at least a factor of 4 larger compared to the significantly spread-out counterion cloud of the pNIPAM microgels studied here.

Importantly, our measurements of the counterion cloud confirm previous results suggesting that the counterion cloud of pNIPAM microgels is the key to their spontaneous deswelling at high microgel concentrations[10,12] and the associated phase behavior in polydisperse suspensions[11]. This is in good agreement with several observations that microgels deswell at concentrations below random close packing and, therefore, before direct contact between neighboring particles[16,26–28]. This observation is not captured by the Hertzian interaction[29–31], the soft-sphere potential[32–35], or a brush-like interaction between the fuzzy coronae[36–38] that have been used to model the behavior of soft colloids. The spontaneous deswelling can explain why the rich phase diagrams predicted with simulations of soft particles[29,39] have not been observed in experiment, where microgels have been found to form close-packed[40,41] or body-centered cubic crystals[4,42], as expected for hard spheres and weakly charged colloids[43].

We also note that the presence of fixed charges and their counterions in pNIPAM microgels was found to be relevant for their swelling behavior in several studies[44–46]. In particular, simulation and dynamic light scattering (DLS) results with pNIPAM microgels are consistent with the charge arrangement we observe in our study[47]. DLS measurements across the volume-phase-transition temperature revealed that the deswelling of the charged shell is slower than that of the microgel core. With the aid of detailed computer simulations, these experimental results were taken as indirect evidence for the presence of fixed charges in the outer fuzzy corona of the microgels.

The screened electrostatic potential due to the charges carried by the microgel and the counterion cloud can be expected to result in an electrostatic microgel-microgel interaction in addition to the interaction of soft, cross-linked pNIPAM spheres in direct contact. In contrast to hard, incompressible colloids with a Yukawa interaction[43], the softness of microgels must play an important role and be at the heart of the shift to higher $\zeta$ of the freezing and melting points. Also, the Percus–Yevick structure factor, $S_{PY}(q)$,[17] has been found to give a good description of the structure of microgel suspensions, albeit with an adapted hard-sphere radius that decreases with concentration[11,16], suggesting that charges can be accounted for in this way. In contrast, $S_{PY}(q)$ cannot describe the suspension structure of other charged colloids[48] or micelles[49], as the electrostatic interaction has a more direct effect on the suspension structure. The open structure and the softness of microgels allow explaining this difference. When the osmotic pressure due to the counterions increases and becomes comparable to the bulk modulus of the microgels, spontaneous deswelling occurs[10,12]. With increasing concentration, the surface-to-surface distance of nearest neighbors, therefore, does not decrease as fast as for incompressible particles. This leaves more screening volume and reduces the effect of electrostatic interactions on the suspension structure. Furthermore, due to the distribution of the fixed charges in the fuzzy corona, about half of the counterions are located inside the microgel, where they screen these fixed charges, lowering the effective charge of the microgel and weakening the electrostatic interaction between particles[50]. It is, therefore, the structure of the counterion cloud that sets the osmotic pressure, triggers the deswelling of the microgels above a critical concentration, and causes the freezing and melting point of the suspension to shift to higher $\zeta$. Microgel deswelling is certainly the most important effect of the counterion cloud with consequences for the phase behavior. In addition, the cloud structure determines the screened electrostatic interaction between microgels that is expected to support the colloidal stabilization of the microgels and to have an effect on the phase behavior at high-volume fractions.

Although the charge density in the periphery of pNIPAM microgels is low, these charges and the corresponding counterions play a decisive role in setting the particle size, consequently affecting interparticle interactions and the suspension phase behavior. Probing its existence and quantifying its properties is thus important, albeit challenging. We have shown that SANS can exploit the contrast difference between different elements or isotopes to resolve the weak signal due to the counterion cloud. We have provided the first measurement of this cloud and showed that it indeed locates in the fuzzy periphery of the microgels. Since contributions to osmotic equilibrium due to polymer-solvent mixing and network elasticity are largely independent of suspension concentration[44], ionic effects turn out to be key to understand the suspension behavior; while at low particle densities, free ions can wander inside and outside the microgels without causing a pressure difference between the inside and outside, at high particle densities, the percolation of counterion clouds can cause the outside osmotic pressure to increase and to induce deswelling. As a result, any model or simulation aiming to capture the phase behavior and mechanics of nearly all microgel suspensions, which are based on either network-charged or peripherically-charged microgels, should explicitly consider the effect of ions. Our work confirms the existence of this cloud of counterions in pNIPAM microgels, which are generally deemed neutral and have been approached by neglecting their peripheric charge. Future theoretical and simulation work should thus consider this fact.

## Methods

### Sample preparation and characterization

The poly-$N$-isopropylacrylamide (pNIPAM) microgels of three different sizes are used as obtained from the group of Prof. L. A. Lyon (Chapman University, USA). The hydrodynamic radii in the fully swollen state at $T = 20\,°C$, determined using DLS, are $R_{sw} = (87.0 \pm 1.7)$ nm (s1), $R_{sw} = (125 \pm 2.7)$ nm (s2), and $R_{sw} = (140 \pm 2)$ nm (s3) Supplementary Information. The collapsed radii, $R_{coll}$, of the microgels are determined using viscometry with dilute microgel suspensions with the aid of the Einstein–Batchelor equation for the relative viscosity Supplementary Information[51]. These characterization measurements allow obtaining the swelling ratio

$$k = \frac{\rho_{solvent}}{\rho_{pNIPAM}} \frac{R_{sw}^3}{R_{coll}^3} \tag{3}$$

used to obtain the effective volume fraction

$$\zeta = k \frac{m_{\text{pNIPAM}}}{m_{\text{tot}}}. \qquad (4)$$

## Dialysis

To extract key parameters of the counterion cloud, suspensions that differ in the counterion type but are otherwise identical are prepared for SANS measurements. The two counterions, $NH_4^+$ and $Na^+$, present after the synthesis, are used for this purpose. Using dialysis, we exchange the counterions to obtain one suspension with $NH_4^+$ and another with $Na^+$ counterions, see Supplementary Information. We first dialyze the microgel suspension, contained inside a dialysis bag, against a NaCl or $NH_4$Cl solution at a concentration of $(167 \pm 5)$ mM for 5 days, exchanging the salt solution once a day. The resultant suspension then contains either $Na^+$ or $NH_4^+$ and $Cl^-$ ions. We then remove the excess salt by dialyzing against ultrapure $H_2O$, again for 5 days and exchanging the $H_2O$ bath daily. The resultant suspensions with either $NH_4^+$ or $Na^+$ counterions are freeze-dried and resuspended in $D_2O$ at a fixed $\zeta$. These suspensions only differ in the type of counterion and show the same suspension behavior, as the counterion concentration is too low to cause significant differences due to the Hofmeister effects as explained in the Supplementary Information.

## Small-angle neutron scattering (SANS)

SANS was performed with dilute, $0.1 \lesssim \zeta \lesssim 0.2$, and more concentrated suspensions, $0.45 \lesssim \zeta \lesssim 0.58$, on the instruments SANS-I[52] and SANS-II[53] of the SINQ neutron source at Paul Scherrer Institut, Switzerland[54]. The microgel suspensions were measured in quartz-glass cuvettes at fixed temperatures of $T = (20 \pm 0.5)$ °C and $(45 \pm 0.5)$ °C. The instruments were configured to cover the $q$ range from about 0.025 nm$^{-1}$ to 0.35 nm$^{-1}$, corresponding to length scales between 30 nm and 600 nm in real space. Both instruments have position-sensitive area detectors covering about one order of magnitude in momentum transfer $q = \frac{4\pi}{\lambda} \sin(\theta/2)$, with $\lambda$ the wavelength, and $\theta$ the scattering angle. The 2D raw data were corrected for dark counts due to electronic noise and stray neutrons and for the sample background originating from the quartz-glass sample container and solvent scattering. The corrected image was calibrated for detector efficiency with an $H_2O$ measurement and was azimuthally averaged to obtain the SANS scattering signal $I(q)$[55].

## Data availability

The SANS scattering curves generated in this study are available in the Zenodo database under the accession code https://doi.org/10.5281/zenodo.7137207. The data analysis and results based on these scattering curves are presented in the main text and in the Supplementary Information. All other data were available from the corresponding author upon request.

## Code availability

The data analysis procedures underlying this article are described in the main text and the Supplementary Information.

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

## Acknowledgements

We acknowledge financial support from the Swiss National Science Foundation (200020_184839) and MCIN/AEI/10.13039/501100011033/FEDER, UE (Grant No. PID2021-122369NB-100). SANS data were taken on the instruments SANS-I and SANS-II at SINQ, Paul Scherrer Institut.

## Author contributions

B.Z. prepared samples, performed measurements, analyzed the data, and wrote the paper. U.G. designed the research, supported and carried out measurements and sample preparation, analyzed data, and wrote the paper. A.F.-N. designed the research, supported measurements and data analysis, and wrote the paper.

## Competing interests

The authors declare no competing interests.
