## [Peer Review File · Nature Communications]

Measuring the counterion cloud of soft microgels using SANS with contrast variationReviewers' Comments:

Reviewer #1:

Remarks to the Author:

Zhou et al. use contrast variation in small angle neutron scattering in a very clever way to measure the counterion distribution around poly(N-isopropylacrylamide) microgels. To my knowledge this is the first work doing this. The ion cloud around the particles is of utmost importance for the particle interaction and colloidal phase behavior. Hence, I think this is an important contribution in the field of microgel research and soft colloids in general. Moreover, it is also an interesting work in terms of method development in the domain of neutron scattering.

I really liked this work and strongly recommend acceptance after very minor revision.

The following points should be revised:

-I think it is somehow miss-leading to talk about SO_3^- instead of SO_4^{2-} . Formally I think we have a sulfuric acid monoester. In case SO_3^- a chemist would assume a C-S bond. The 4th oxygen also contributes to the polarity of the group. However, the reason for talking about SO_3^- is explained in the SI. Nevertheless, in all other works I know the authors talk about surface sulfate groups.

- page 3, line 136: Maybe I missed this point, but did the authors determine the residual light water content in the freeze dried samples? Is there any? Does this have an impact on the results?

-Page 5, line 198: Here, the authors claim that the polymer contribution does not depend on counterion type. Maybe some more explanation would be necessary. Can Hofmeister effects really be ruled completely?

Some typos:

line 035: ...can spontaneously....

line 096: ...utmost...

line 123: N, N' -methylene-

line 246: Fig.2 (A) Subtraction....

line 256-257: ...the contrast...

Reviewer #2:

Remarks to the Author:

This is a strong contribution that I believe is worthy of publication in Nature Communications. The most noteworthy aspect of the work is the direct measurement of the counterion cloud surrounding polymeric microgel particles, using SANS and taking advantage of different scattering length densities of different counterions. This is very exciting and will make an important contribution to the literature; the approach developed by the authors could also be applied towards other types of charged colloids, opening up the potential for further study and improved understanding of colloidal materials. While the presence of the counterion cloud has been theorized for some time, to my knowledge the authors are correct that there have been no direct measurements for polymeric microgel particles. I would have guessed that the cloud would be too diffuse and not of sufficient density to be measurable by SANS, but the data appear to be of very good quality, and the data analysis is sound. The authors also explain some implications of their results in understanding the phase behavior of microgel particles. Overall, I think this is a significant contribution to the literature. The paper itself is well-written and clear. I recommend publication with no further revision.

Answers to the reviewers

April 11, 2023

I. Reviewer 1

Zhou et al. use contrast variation in small angle neutron scattering in a very clever way to measure the counterion distribution around poly(N-isopropylacrylamide) microgels. To my knowledge this is the first work doing this. The ion cloud around the particles is of utmost importance for the particle interaction and colloidal phase behavior. Hence, I think this is an important contribution in the field of microgel research and soft colloids in general. Moreover, it is also an interesting work in terms of method development in the domain of neutron scattering. I really liked this work and strongly recommend acceptance after very minor revision. The following points should be revised:

-I think it is somehow miss-leading to talk about SO₃⁻ instead of SO₄⁻. Formally I think we have a sulfuric acid monoester. In case SO₃⁻ a chemist would assume a C-S bond. The 4th oxygen also contributes to the polarity of the group. However, the reason for talking about SO₃⁻ is explained in the SI. Nevertheless, in all other works I know the authors talk about surface sulfate groups.

The charged groups at the end of the pNIPAM chains are due to the APS initiator and are -CH₂-O-SO₂-O⁻. We completely agree with the referee in that four oxygens are present and that, as a result, one could refer to SO₄⁻ groups. The charge is nevertheless carried by the SO₃ part, which is connected to the pNIPAM chain by the fourth oxygen atom. To more clearly refer to the group responsible for the charge, we refer to -CH₂-O-SO₂-O⁻ groups due to APS, both in the main text and in the supporting information (SI).

- page 3, line 136: Maybe I missed this point, but did the authors determine the residual light water content in the freeze dried samples? Is there any? Does this have an impact on the results?

The freeze-dried pNIPAM powder was not tested for residual light water (H₂O). The presence of residual H₂O cannot be excluded and has been estimated to be close to 1

H₂O molecule per NIPAM monomer in freeze-dried pNIPAM [T. Kyrey et al., PCCP 23, 14252 (2021)]. However, after resuspending the microgels in D₂O, the contamination with H₂O is at most on the 1 wt% level and implies a very small reduction of the scattering-length density of the solvent. The contrast of pNIPAM and the counterions is reduced by < 1%, which is irrelevant for our data analysis and the conclusions. We now mention this at the end in section 1 of the SI.

-Page 5, line 198: Here, the authors claim that the polymer contribution does not depend on counter ion type. Maybe some more explanation would be necessary. Can Hofmeister effects really be ruled completely?

The ion concentrations in the suspensions we use here are too low for Hofmeister effects to play a relevant role, and our results support this. Indeed, the scattering contrast obtained between sample and solvent can be explained by considering the involved scattering length densities, without the need to consider a change in the microgel structure. In addition, such a change, if present, would not be able to describe all the relevant features in our data, as detailed in sections 4 and 5 of the SI. From prior published work, and as explained in section 5 of the SI, the Na⁺ and NH₄⁺ concentrations would need to be about an order of magnitude higher for us to observe a difference in the microgel swelling behavior due to Hofmeister effects. We refer to the SI in connection to this in line 200 in the main text.

II. Reviewer 2

This is a strong contribution that I believe is worthy of publication in Nature Communications. The most noteworthy aspect of the work is the direct measurement of the counterion cloud surrounding polymeric microgel particles, using SANS and taking advantage of different scattering length densities of different counterions. This is very exciting and will make an important contribution to the literature; the approach developed by the authors could also be applied towards other types of charged colloids, opening up the potential for further study and improved understanding of colloidal materials. While the presence of the counterion cloud has been theorized for some time, to my knowledge the authors are correct that there have been no direct measurements for polymeric microgel particles. I would have guessed that the cloud would be too diffuse and not of sufficient density to be measurable by SANS, but the data appear to be of very good quality, and the data analysis is sound. The authors also explain some implications of their results in understanding the phase behavior of microgel particles. Overall, I think this is a significant contribution to the literature. The paper itself is well-written and clear. I recommend publication with no further revision.

We thank the review for the very positive assessment!

Reviewers' Comments:

Reviewer #1:

Remarks to the Author:

The authors have addressed all points which I mentioned in my report and the manuscript has been improved accordingly.

As already pointed out in my first report, I think this is a very nice work and it can now be published as is.

Answers to the reviewers

June 9, 2023

We have obtained only one short comment from Reviewer 1:

I. Reviewer 1

The authors have addressed all points which I mentioned in my report and the manuscript has been improved accordingly. As already pointed out in my first report, I think this is a very nice work and it can now be published as is.

We thank the review for the very positive assessment!